# Drastic changes in conformational dynamics of the antiterminator M2-1 regulate transcription efficiency in *Pneumovirinae*

Cedric Leyrat[1], Max Renner[1], Karl Harlos[1], Juha T Huiskonen[1], Jonathan M Grimes[1,2]*

[1]Division of Structural Biology, Wellcome Trust Centre for Human Genetics, Oxford, United Kingdom; [2]Diamond Light Source Ltd, Didcot, United Kingdom

**Abstract** The M2-1 protein of human metapneumovirus (HMPV) is a zinc-binding transcription antiterminator which is highly conserved among pneumoviruses. We report the structure of tetrameric HMPV M2-1. Each protomer features a N-terminal zinc finger domain and an α-helical tetramerization motif forming a rigid unit, followed by a flexible linker and an α-helical core domain. The tetramer is asymmetric, three of the protomers exhibiting a closed conformation, and one an open conformation. Molecular dynamics simulations and SAXS demonstrate a dynamic equilibrium between open and closed conformations in solution. Structures of adenosine monophosphate- and DNA- bound M2-1 establish the role of the zinc finger domain in base-specific recognition of RNA. Binding to 'gene end' RNA sequences stabilized the closed conformation of M2-1 leading to a drastic shift in the conformational landscape of M2-1. We propose a model for recognition of gene end signals and discuss the implications of these findings for transcriptional regulation in pneumoviruses.

**\*For correspondence:**
jonathan@strubi.ox.ac.uk

**Competing interests:** The authors declare that no competing interests exist.

**Reviewing editor**: Volker Dötsch, Goethe University, Germany

## Introduction

Human metapneumovirus (HMPV) is a negative-strand, non-segmented ssRNA virus of the *Paramyxoviridae* family and a major cause of acute respiratory tract infections in children, elderly and immunocompromised populations worldwide (***van den Hoogen et al., 2002***; ***van den Hoogen et al., 2003***). HMPV and the closely-related respiratory syncytial virus (RSV) constitute respectively the *Metapneumovirus* and *Pneumovirus* genera of the *Pneumivirinae* subfamily. These viruses share conserved replication strategies and similar genome organizations with other members of the *Mononegavirales* order, which includes numerous important human pathogens such as measles, rabies, and Ebola virus. The HMPV genome encodes nine proteins, three of which are necessary and sufficient for viral replication, namely the nucleoprotein (N), the RNA-dependent RNA polymerase (L) and its essential cofactor, the phosphoprotein (P). The genomic RNA is encapsidated by the N protein, which acts as a template for the L protein that is responsible for both replication and transcription. The replicase is highly processive, and generates a complete, encapsidated positive-sense antigenome, which is in turn used as a template for the synthesis of genomic RNA. The transcriptase produces capped and polyadenylated monocistronic mRNAs using a sequential stop and restart mechanism in which the polymerase responds to *cis*-acting signals located in intergenic regions (***Sutherland et al., 2001***).

Members of the *Pneumovirinae* subfamily harbour highly conserved 9–10 nucleotide transcription promoters ('gene start') and semi-conserved 12–13 nucleotide 'gene end' (GE) signals with the consensus sequence 5'-AGUUAnnnAAAAA-3' (positive sense), which direct polyadenylation and release

**eLife digest** To produce a protein from a gene, the gene must first be transcribed to make a molecule of RNA. In general, the enzyme building the RNA molecule stops building when it reaches the end of a gene and encounters a termination signal. When a virus replicates, however, it needs to transcribe all the genes in its genome, so it relies on antiterminator proteins to make the enzyme building the RNA ignore the termination signal. Therefore, medicines that stop antiterminators working could stop viral infections spreading.

Human metapneumovirus (HMPV) can cause severe respiratory infections in children, the elderly and people with weakened immune systems. A protein called M2-1 that is found inside HMPV must be present for the virus to infect humans, and it was recently shown that this protein plays a role in antitermination in a virus closely related to HMPV.

Using a range of techniques, including X-ray crystallography and molecular dynamics simulations, Leyrat et al. worked out the structure of M2-1 in HMPV, and showed that it can flip between 'open' and 'closed' forms. The open structure presents surfaces that could be targeted by antiviral drugs. When M2-1 binds to RNA, the closed structure is stabilized as a result of the RNA binding to two separate sites on the protein.

Leyrat et al. suggest that similar antiterminator proteins in related viruses—including respiratory syncytial virus, Marburg and Ebola—could also bind in this way. Leyrat et al. also propose a model describing how M2-1 can recognize the end of a gene, which could help with the development of new antiviral treatments.

of nascent mRNA and are critical for polymerase processivity (**Harmon et al., 2001**; **Sutherland et al., 2001**). Termination of each gene is required to allow transcription of the next gene downstream, and the propensity of the polymerase to dissociate from its template results in transcriptional attenuation at each gene junction (**Fearns and Collins, 1999**).

HMPV M2-1 is a basic protein of 187 amino acids that is required for virus infectivity in vivo, but has been found to be dispensable for recovery or growth of recombinant virus in tissue culture (**Buchholz et al., 2005**). The closely-related M2-1 protein from avian metapneumovirus, which shares 85% sequence identity with HMPV M2-1, has been shown to increase minigenome expression by at least 100-fold (**Naylor et al., 2004**). These observations are in contrast with studies of RSV M2-1, which was shown to be absolutely essential for transcription of full-length viral mRNAs (**Collins et al., 1996**; **Fearns and Collins, 1999**). M2-1 proteins from HMPV and RSV are amongst the most conserved within the *Pneumivirinae* and share 38% overall sequence identity, suggesting similar functional roles. In addition, M2-1 shares structural and functional similarity with Ebola virus VP30 (**Blondot et al., 2012**). Recently, the crystal structure of RSV M2-1 has been solved by X-ray crystallography, revealing a tight, disk-like tetrameric assembly (**Tanner et al., 2014**), which contrasts with previous solution studies that suggested a non-globular, extended tetramer (**Esperante et al., 2011**).

M2-1 is recruited to the viral transcription complex by the intrinsically-disordered P protein (**Khattar et al., 2001**; **Derdowski et al., 2008**). The recruitment of M2-1 occurs through an interaction with the M2-1 core domain (**Tran et al., 2009**; **Blondot et al., 2012**), resulting in the formation of a high-affinity, non-globular complex (**Esperante et al., 2012**), which in RSV is controlled by phosphorylation of Thr108 of the P protein (**Asenjo et al., 2006**). Studies on RSV have shown that M2-1 functions as both an intragenic and intergenic transcription antitermination factor in *Pneumoviruses*, allowing synthesis of complete viral mRNAs and inhibiting transcription termination at the GE signal with various efficiencies, resulting in an increased proportion of polycistronic read-through mRNAs (**Hardy and Wertz, 1998**; **Hardy et al., 1999**). Interestingly, NMR studies have shown that the RSV M2-1 core domain preferentially recognizes poly-A tails of viral mRNAs (**Blondot et al., 2012**), and this property has been confirmed with full-length protein using fluorescence anisotropy (**Tanner et al., 2014**). On the basis of this data, it has been suggested that M2-1 likely binds nascent mRNA transcripts, thus preventing premature termination through stabilization of the transcription complex and inhibition of RNA secondary structure formation (**Blondot et al., 2012**; **Tanner et al., 2014**). As structural data are lacking for HMPV M2-1, its role in HMPV transcription antitermination has remained elusive.

In this study, we report X-ray crystallographic structures of the HMPV M2-1 at resolutions ranging from 2.0 to 2.5 Å. Unlike the disk-like assembly reported for RSV, our HMPV M2-1 structures revealed dissociation of one protomer core domain from the tetramer interface. Solution small angle X-ray scattering (SAXS) and atomistic coarse-grained molecular dynamics (MD) simulations demonstrated that M2-1 behaves as a dynamic, modular protein in equilibrium between open and closed forms. Crystallographic studies of M2-1 bound to nucleic acids reveal how the M2-1 zinc finger specifically recognizes RNA. Finally, SAXS and electron microscopy showed that interaction with GE signals induces the closed conformation of M2-1, a process which is coupled with concentration-dependent aggregation in solution. Our results provide a structural basis for the recognition of GE signals by M2-1, and the prevention of premature mRNA termination.

## Results

### HMPV M2-1 forms a tetramer with an open and a closed conformation

M2-1 crystallized in space group $P2_1$ and its structure was solved at 2.5 Å resolution, by multiwavelength anomalous dispersion (MAD) using zinc anomalous scattering (*Table 1*). The structure reveals an asymmetric tetramer, where three monomers display a canonical globular conformation, whilst unexpectedly, the fourth protomer adopts an open conformation where the core domain has flipped away from the rest of the molecule by approximately 60 Å (*Figure 1A,C*). This open conformation is in

**Table 1.** X-ray data collection and refinement statistics

| Data Set | MAD–Peak | MAD–High-Energy Remote | Native | AMP-Soak | DNA-Soak |
|---|---|---|---|---|---|
| Data collection and processing statistics | | | | | |
| λ (Å) | 1.2828 | 1.2802 | 0.9686 | 0.9686 | 0.9795 |
| Resolution range (Å) | 33.13–2.47 | 33.13–2.51 | 44.81–2.10 | 49.80–2.01 | 63.04–2.28 |
| Space group | $P2_1$ | $P2_1$ | $P2_1$ | $P2_1$ | $P2_1$ |
| Unit cell constants (Å)/(°) | 50.2, 92.7, 82.8/90.0, 94.5, 90.0 | | 50.0, 93.4, 85.2/90.0, 95.4, 90.0 | 50.1, 93.6, 85.6/90.0, 95.8, 90.0 | 50.1, 93.9, 85.5/90.0, 95.8, 90.0 |
| Measured Reflections | 182, 655 | 167, 780 | 538, 740 | 660, 719 | 235, 027 |
| Unique Reflections | 27, 190 | 25, 798 | 45, 494 | 51, 659 | 35, 938 |
| Completeness (%) (outer shell) | 99.8 (99.9) | 99.9 (99.7) | 99.8 (99.2) | 98.8 (85.1) | 99.7 (98.8) |
| Multiplicity (%) (outer shell) | 6.7 (6.7) | 6.5 (3.3) | 11.8 (5.8) | 12.8 (5.1) | 6.5 (5.0) |
| $R_{pim}$ (outer shell) | 0.039 (0.604) | 0.031 (0.367) | 0.021 (0.420) | 0.019 (0.401) | 0.023 (0.421) |
| $R_{merge}$ (outer shell) | 0.078 (1.338) | 0.062 (0.516) | 0.074 (0.937) | 0.068 (0.839) | 0.054 (0.843) |
| Mean (<I>/sd <I>) (outer shell) | 12.2 (1.7) | 14.9 (1.9) | 23.1 (2.0) | 22.5 (1.8) | 19.6 (1.8) |
| Refinement and Ramachandran statistics | | | | | |
| $R_{work}$ (%) | 23.34 | | 19.23 | 18.64 | 19.35 |
| $R_{free}$ (%) | 26.15 | | 22.20 | 20.83 | 22.21 |
| RMSD Bond lengths (Å) | 0.012 | | 0.009 | 0.010 | 0.009 |
| RMSD Bond angles (°) | 1.18 | | 1.00 | 0.96 | 1.03 |
| Residues in preferred regions (%) | 95.3 | | 97.8 | 97.3 | 97.3 |
| Residues in allowed regions (%) | 3.9 | | 2.2 | 2.4 | 2.4 |
| Outliers (%) | 0.8 | | 0.0 | 0.3 | 0.3 |
| PDB ID | 4CS7 | | 4CS8 | 4CS9 | 4CSA |

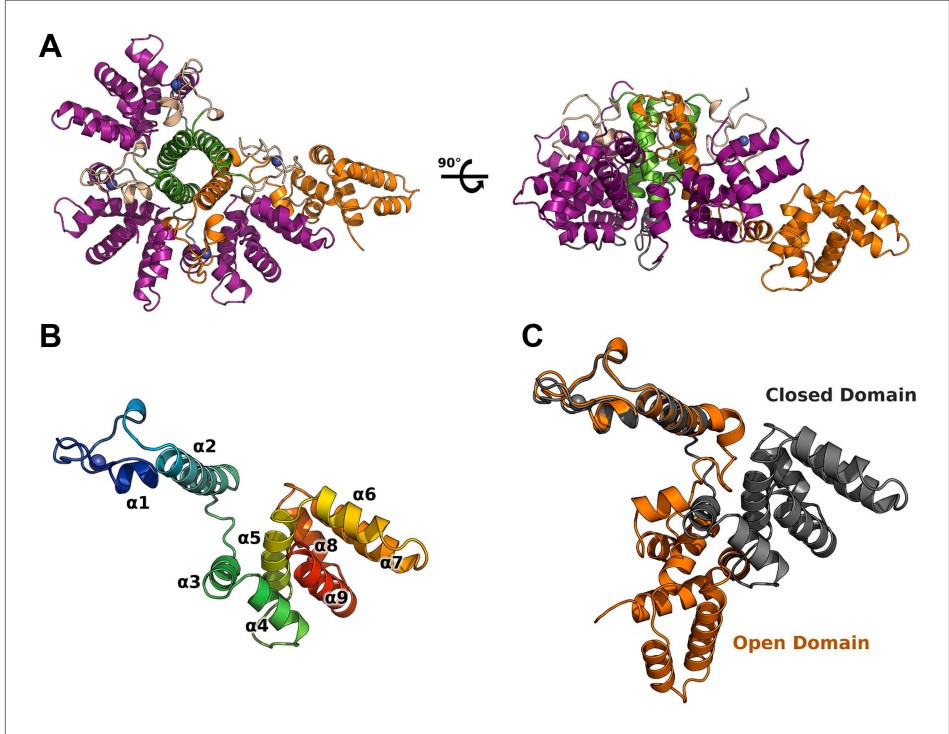

**Figure 1**. Crystal structure of HMPV M2-1. (**A**) Top view (left) and side view (right) of M2-1 asymmetric tetramer in cartoon representation. Closed protomers are coloured by domain with the zinc finger in wheat, tetramerization helix in green, and core domain in purple. Zinc atoms are represented as blue spheres. The open protomer is coloured in orange. (**B**) Close-up view of a closed protomer. The molecule is coloured from blue (N terminus) to red (C terminus) and secondary structure elements are labelled. (**C**) Comparison between open (orange) and closed protomer (grey).

The following figure supplements are available for figure 1:

**Figure supplement 1**. Flexibility, interfaces details and packing of M2-1 in the crystalline state.

**Figure supplement 2**. Classical molecular dynamics simulations of the closed and open state models of M2-1.

striking contrast with the symmetrical RSV M2-1 tetramer (*Tanner et al., 2014*). Subsequently, a second structure solved by molecular replacement at 2.1 Å resolution showed significant interdomain motions (*Figure 1—figure supplement 1A*), in addition to highlighting the flexibility of the molecule and the importance of crystallographic packing for the stability of the assembly (*Figure 1—figure supplement 1F*). In the open conformer, the core domain, which has dissociated from the tetramer, packs against closed subunits of crystallographically related molecules, resulting in the formation of a planar two-dimensional array (*Figure 1—figure supplement 1F*).

## HMPV M2-1 protomers exhibit a modular architecture

Each M2-1 monomer is characterized by a CCCH zinc finger (residues 1 to 30), rigidly anchored to a tetramerization helix (residues 31 to 52) through hydrophobic interactions. A flexible linker (residues 53 to 65) connects to the core domain (residues 66 to 167), which adopts a globular, α-helical fold (*Figure 1B*). Residues 168 to 187 form an intrinsically disordered C-terminal extension, however partial ordering is observed for residues 168 to 175 in one of the closed protomers.

The CCCH zinc finger adopts a fold that is similar to known eukaryotic motifs with low secondary structure content (*Guo et al., 2004*; *Hudson et al., 2004*). The zinc atom is coordinated by Cys7, Cys15, Cys21, and His25 with canonical tetrahedral geometry (*Figure 1—figure supplement 1C*). The zinc finger interacts with the tetramerization helix through a hydrophobic interface that involves Pro6, Val11, Tyr27, Trp30, Tyr34, and Leu35 (*Figure 1—figure supplement 1D*). The tetramerization helical

interface is mainly stabilized through hydrophobic contacts between Leu35, Leu36, Ile37, Leu42, Leu43, Leu46, and Leu47 of related monomers, and a salt bridge between Asp32 and Arg33.

In contrast to the zinc finger and tetramerization interfaces, the C-terminal core domain interacts with the zinc finger, tetramerization and core domains of adjacent protomers mostly through polar contacts. In particular, Glu109 forms a salt bridge with Arg12, and Gln104 and Gln106 share hydrogen bonds with Asn16, Asn44, and Gln45 side chains (*Figure 1—figure supplement 1E*). Additional stability is provided by a few hydrophobic interactions and two salt bridges between core domains of neighbouring protomers. Comparisons between crystallographic monomers in the closed state reveal movements of the rigid core domain relative to the tetramerisation helix and zinc finger (RMSD of 0.9–1.1 Å on Cα atoms of the core domain), suggesting that M2-1 displays considerable flexibility even in the closed state (*Figure 1—figure supplement 1B*).

## Molecular dynamics simulations of the tetramer in the open and closed states reveal a rigid tetramerization domain and a dynamic core domain

On the basis of the crystal structure, we used computational modelling to construct molecular models of the fully closed and fully open states, in which all protomers are dissociated from the tetramerization domain. We investigated the stability of both forms of M2-1 using classical molecular dynamics simulations (MDS), and calculated root mean square fluctuations (RMSF) along the trajectories (*Figure 1—figure supplement 2*). The fully open state (*Figure 1—figure supplement 2B*) displays high levels of motion in the whole protein, with only the tetramerization helix remaining rigid (*Figure 1—figure supplement 2D*), indicating it is the most stable region of the protein. In contrast, the closed state displays much lower flexibility due to stabilization of the protein through intramolecular contacts between the core domain and the rest of the molecule (*Figure 1—figure supplement 2A*). In particular, the zinc finger domains and tetramerization helices appear very rigid, while some mobility is retained in the flexible loop connecting the tetramerization and core domains (*Figure 1—figure supplement 2C*). Notably, the rigid core domain can flex with respect to the zinc finger domains and tetramerization helices, consistent with observations made from the crystal structures (*Figure 1—figure supplement 1B*), which can be linked to a rather dynamic hydrogen bonding network between the core domain and the rest of the molecule (*Figure 1—figure supplement 2E*).

## Effects of solvent composition on the stability and behaviour of M2-1 in solution

To gain additional insights into the behaviour of M2-1, we turned to SAXS. Initial attempts to measure M2-1 in low salt conditions resulted in precipitation (data not shown), whilst the formation of higher order oligomers or soluble aggregates occurred in the presence of 600 mM, 1.15 M and 3 M NaCl (*Table 2*). The formation of soluble aggregates was reversible upon dilution of M2-1 in buffer containing non-detergent sulfobetaines (NDSBs). Previous work has shown that NDSBs can improve protein (re) folding (*Expert-Bezancon et al., 2003*) and induce compaction of intrinsically disordered regions (*Leyrat et al., 2013*). These observations were consistent with increases in thermal stability of +6.3 and +10.0° upon addition of 500 mM NDSB or 1 M NaCl to M2-1 (*Figure 2—figure supplement 1A,B*).

SAXS characterization of M2-1 in the presence of 300 mM NaCl and 1 M NDSB, or 150 mM NaCl and 500 mM NDSB resulted in $R_g$ values of 3.9 and 3.5 nm, respectively (*Table 2*), which were larger than the value of 3.3 nm calculated for the closed state model from MDS. Molecular weights of 82–93 kDa were estimated based on the volume-of-correlation invariant (*Rambo and Tainer, 2013*), and were consistent with the theoretical value of 91.8 kDa expected for a tetramer.

## SAXS and ensemble optimisation reveal the equilibrium of M2-1 between the open and closed states in solution

To study the conformational dynamics of M2-1 in solution, we generated multiple ensembles of M2-1 models by using all atom coarse-grained MD simulations and performed ensemble optimization against the SAXS data. The use of conformational ensembles of M2-1 enabled a detailed understanding of the transition between open and closed states (*Figure 2A*) and allowed the quantification of the relative populations of conformers (*Figure 2I*).

SAXS data measured in conditions in which M2-1 is in a tetrameric form were generally well-fitted by our ensemble optimization approach (*Figure 2H*, *Figure 2—figure supplement 1G*) with $\chi_{exp}$ values of less than one (*Table 2*). *Figure 2B–G* shows the calculated conformational landscapes along the radius of gyration ($R_g$) and maximal intramolecular distance ($D_{max}$) in various buffer environments.

**Table 2.** SAXS-derived parameters

| Buffer conditions | c (mg/ml) | MW (kDa) | Rg (nm) | Dmax (nm) | $\chi_{exp}$ |
|---|---|---|---|---|---|
| 20 mM Tris pH 7.5 300 mM NaCl 1 M NDSB-201 | 2.00 | 93 | 4.09 | 13.71 | 0.983 |
| – | 1.50 | 89 | 3.90 | 13.22 | 0.908 |
| – | 1.00 | 82 | 3.84 | 13.01 | 0.857 |
| 20 mM Tris pH 7.5 150 mM NaCl 500 mM NDSB-201 | 0.75 | 87 | 3.52 | 12.60 | 0.839 |
| 20 mM Tris pH 7.5 150 mM NaCl 500 mM NDSB-201 + 1 M Gdn-HCl | 0.50 | 124 | 4.24 | 14.50 | 0.829 |
| 20 mM Tris pH 7.5 150 mM NaCl 500 mM NDSB-201 + 2 M Gdn-HCl | 0.50 | 140 | 4.72 | 15.34 | 0.856 |
| 20 mM Tris pH 7.5 150 mM NaCl 500 mM NDSB-201 + 3 M Gdn-HCl | 0.50 | 88 | 5.24 | 18.35 | 0.893 |
| 20 mM Tris pH 7.5 1.15 M NaCl 250 mM AMP | 1.80 | 85 | 4.30 | 14.70 | 0.986 |
| 20 mM Tris pH 7.5 150 mM NaCl 250 mM AMP | 2.00 | 97/78* | 4.01 | 13.26 | >10 |
| 20 mM Tris pH 7.5 150 mM NaCl 500 mM NDSB-201 250 mM AMP | 0.75 | 90 | 3.91 | 13.70 | 0.814 |
| 20 mM Tris pH 7.5 150 mM NaCl 500 mM NDSB-201 250 mM UMP | 0.75 | 92 | 3.86 | 13.50 | 0.843 |
| 20 mM Tris pH 7.5 150 mM NaCl 500 mM NDSB-201 250 mM CMP | 0.75 | 87 | 3.99 | 13.98 | 0.808 |
| 20 mM Tris pH 7.5 150 mM NaCl 500 mM NDSB-201 + 5 mM EDTA | 0.75 | 82 | 3.86 | 14.2 | 0.927 |
| 20 mM Tris pH 7.5 575 mM NaCl | 0.4 | 234 | 4.83 | 17.2 | N.D |
| 20 mM Tris pH 7.5 3 M NaCl | 0.4 | 463 | 8.62 | 26.7 | N.D |
| 20 mM Tris pH 7.5 1.15 M NaCl | 0.4 | 178 | 5.77 | 20.10 | N.D |
| 20 mM Tris pH 7.5 1.15 M NaCl | 0.90 | 254 | 5.91 | 20.50 | N.D |
| 20 mM Tris pH 7.5 1.15 M NaCl | 1.50 | 305 | 6.10 | 21.30 | N.D |
| 20 mM Tris pH 7.5 1.15 M NaCl | 3.00 | 580 | 7.20 | 25.20 | N.D |

*97 kDa assuming 90% protein + 10% RNA and 78 kDa assuming only protein.

This analysis revealed a dynamic equilibrium between open and closed states, where the closed state accounted for 50% of the conformers in the presence of 500 mM NDSB, and 40% in 1 M NDSB (*Figure 2A,B,I*). Interestingly, incubation of M2-1 with 5 mM of EDTA decreased the proportion of closed-state conformers from 50% to ~20% (*Figure 2D,I*), suggesting that a bound zinc ion and the integrity of the zinc finger motif is important for the stability of the closed state.

The addition of increasing concentrations of guanidinium hydrochloride (Gdn-HCl) from 1 M to 3 M resulted in the disappearance of closed state conformers (*Figure 2E,F,G*). In 1 M Gdn-HCl, 65% of the conformers are in an intermediate state with two core domains opened, while the addition of 3 M Gdn-HCl induces a shift in the equilibrium such that nearly 60% are found to be in the fully open state (*Figure 2I*). Addition of Gdn-HCl was unlikely to cause unfolding of the core domain or tetramer dissociation, as evidenced by the good quality of the fit using our models (*Figure 2H*). This was despite some discrepancies in the molecular weight estimated from the SAXS data (*Table 2*), which may result from the low signal-to-noise ratio. The ability to shift the equilibrium towards open states with low concentrations of Gdn-HCl is consistent with the polar nature of the interactions between the core domain and the rest of the molecule, and suggests that these interactions are weak in solution. An additional possibility, that cannot be ruled out on the basis of the SAXS data given the low resolution of the technique, is that the disruption of the closed state by Gdn-HCl may be enhanced through destabilization of the zinc finger structure by the denaturant, as this would result in only minor changes in the scattering profile.

The conformational equilibrium of M2-1 quantified by ensemble analysis was corroborated by comparing dimensionless Kratky plots of M2-1 measured in different buffer conditions (*Figure 2—figure supplement 2*), providing a model-free analysis of M2-1 flexibility. The analysis shows that M2-1

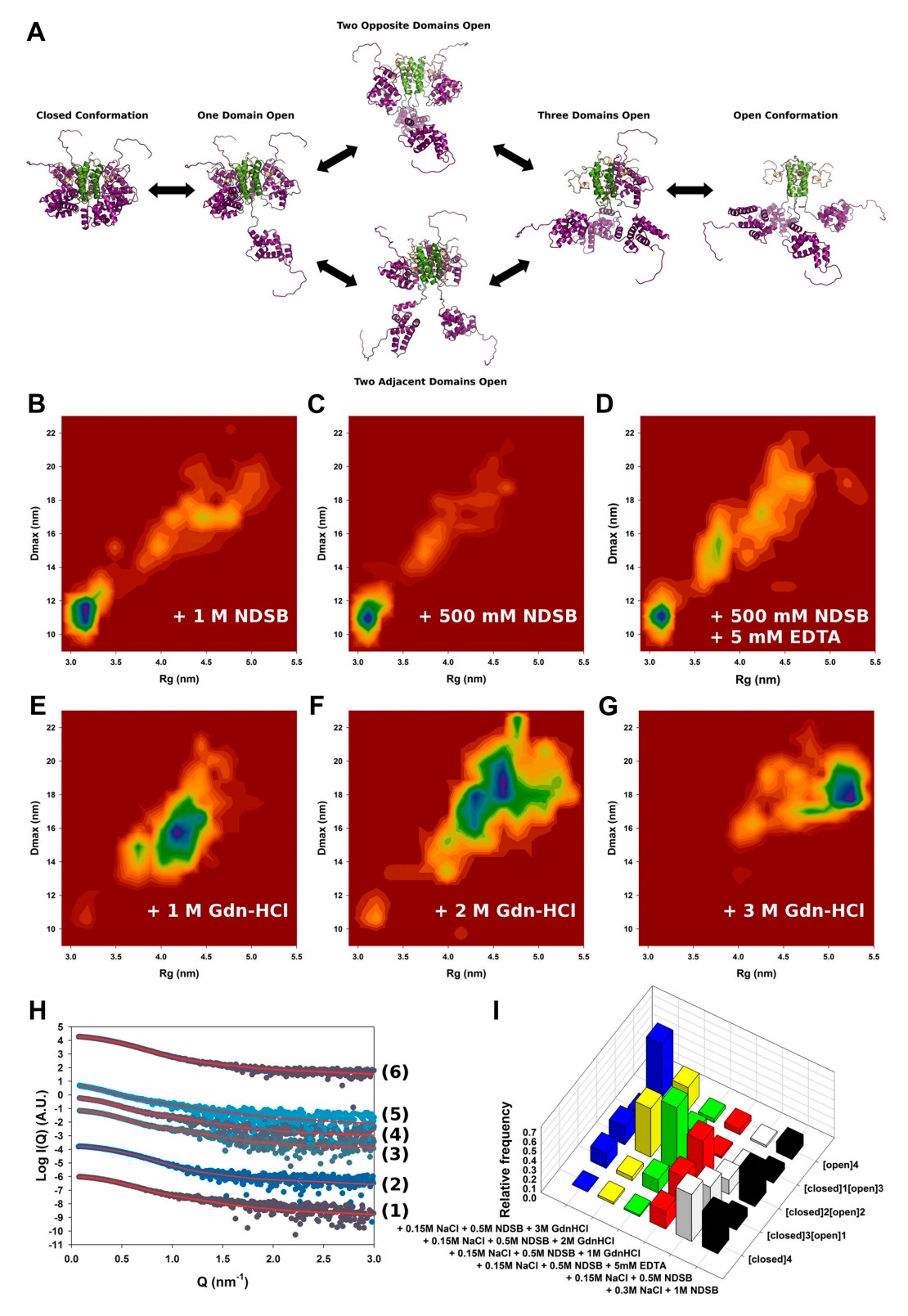

**Figure 2**. Solution structure of HMPV M2-1. (**A**) Model for domain opening and closure. The structures are coloured by domain with the zinc finger in wheat, tetramerization domain in green, and core domain in purple. The N-terminal histag is omitted for clarity. (**B**–**G**) Two-dimensional histogram representations of radius of gyration ($R_g$) and maximal intramolecular distance ($D_{max}$) distributions for M2-1 in the presence of 20 mM Tris pH 7.5 and

*Figure 2. Continued on next page*

*Figure 2. Continued*

300 mM NaCl and 1 M NDSB-201 (**B**), 150 mM NaCl and 500 mM NDSB-201 (**C**), 150 mM NaCl, 500 mM NDSB-201, and 5 mM EDTA (**D**), 150 mM NaCl, 500 mM NDSB-201 and 1 M, 2 M or 3 M guanidinium hydrochloride (**E**, **F** and **G**). (**H**) Fitted SAXS profiles using optimized ensembles of 50 models. From bottom to top, data measured in the presence of 20 mM Tris pH 7.5, 150 mM NaCl, and 500 mM NDSB-201 (1), 150 mM NaCl, 500 mM NDSB-201 and 5 mM EDTA (2), 1 M, 2 M, or 3 M guanidinium hydrochloride (3, 4, 5), or 300 mM NaCl and 1 M NDSB-201 (6). (**I**) Relative frequencies of selection of the conformers shown in **A** in various buffer environments.

The following figure supplements are available for figure 2:

**Figure supplement 1**. Conformational landscape of M2-1 in the presence of nucleotides studied by SAXS.

**Figure supplement 2**. Model-free analysis of M2-1 conformational landscape.

measured in the presence of NDSB-201 is more flexible than a theoretical ensemble of closed state models, and that this flexibility is further increased by addition of Gdn-HCl. As the dimensionless Kratky plots are normalized by the particle's $R_g$ and forward scattering intensity $I_0$, they also enable qualitative analysis of M2-1 conformational states in conditions where a small proportion of higher order oligomers is formed such as in the absence of NDSB. Normalized Kratky plots of data measured in the presence of 600 mM, 1.15 M or 3 M NaCl were relatively similar to the plot obtained in the presence of 500 mM NDSB, indicating a comparable equilibrium between open and closed states, and suggesting that the main effect that NDSB exerts on M2-1 in solution is the disruption of soluble aggregates.

Addition of AMP in the absence of NDSB and in the presence of high salt concentrations resulted in improved solubility and inhibited the formation of soluble aggregates (*Table 2*). The conformational landscape is dominated in these conditions by conformers in a fully open state (45%) (*Figure 2—figure supplement 1C,H*), likely resulting from electrostatic screening. Addition of nucleotides in the presence of 150 mM NaCl and 500 mM NDSB also led to an increase in open state conformers (*Figure 2—figure supplement 1D–F*), as observed in 1.15 M NaCl buffer. Interestingly, measurements performed in the presence of 150 mM NaCl and in the absence of NDSB resulted in data that was consistent with an M2-1 tetramer (*Table 2*), but could not be fitted correctly with our ensemble ($\chi_{exp} > 10$), in particular at high Q (*Figure 2—figure supplement 1G*, bottom curve), suggesting that M2-1 binds AMP in these conditions.

## M2-1 binds more strongly to AMP than to other mononucleotides

To study the recognition of RNA by M2-1, we performed fluorescence-based thermal shift assays (TSA) in the presence of nucleotides (*Figure 3—figure supplement 1*). The effects of increasing concentrations of AMP, UMP, GMP, and CMP on M2-1 melting profiles were measured in low salt conditions (*Figure 3—figure supplement 1A–D*), and differences in $T_m$ and $\Delta\Delta G$ values were consecutively calculated as described previously (*Layton and Hellinga, 2010*; *Figure 3—figure supplement 1E,F*). Interestingly, AMP consistently induced the largest shifts in thermal stability of M2-1 (*Figure 3—figure supplement 1A,E*), suggesting specificity for the adenine moiety. Extracted apparent $K_d$ values were in the millimolar range and indicated that $K_d(AMP) < K_d(UMP) < K_d(GMP) < K_d(CMP)$ (with values of 8, 21, 24, and 35 mM, respectively), an observation which is consistent with the A/U rich nature of the GE RNA sequences, and with binding studies performed on RSV M2-1 that showed highest affinity for poly-A RNA (*Tanner et al., 2014*).

Next, we performed crystal soaking experiments to obtain structural insights into the interaction of M21 with RNA. Crystals soaked with CMP, GMP, or UMP (or UU and GUU) revealed no additional electron density, however addition of AMP led to $F_o$–$F_c$ difference maps with strong density in the vicinity of the zinc finger. This region harbours conserved aromatic and positively charged residues on its surface, as can be seen in a superimposition with the equivalent RSV zinc finger (*Figure 3A*). In three protomers out of four, one AMP molecule is buttressed between the zinc finger and core domain of a crystallographically related molecule, resulting in clearly defined electron density (*Figure 3—figure supplement 2*). The adenine moiety interacts with the zinc finger through stacking interactions with Lys8 and Phe23 and hydrogen bonding to the backbone nitrogen of Lys8, and the carbonyl oxygen of Pro6 (*Figure 3B*) or the sulphur atom of Cys7 (*Figure 3C*), depending on the orientation of the bound

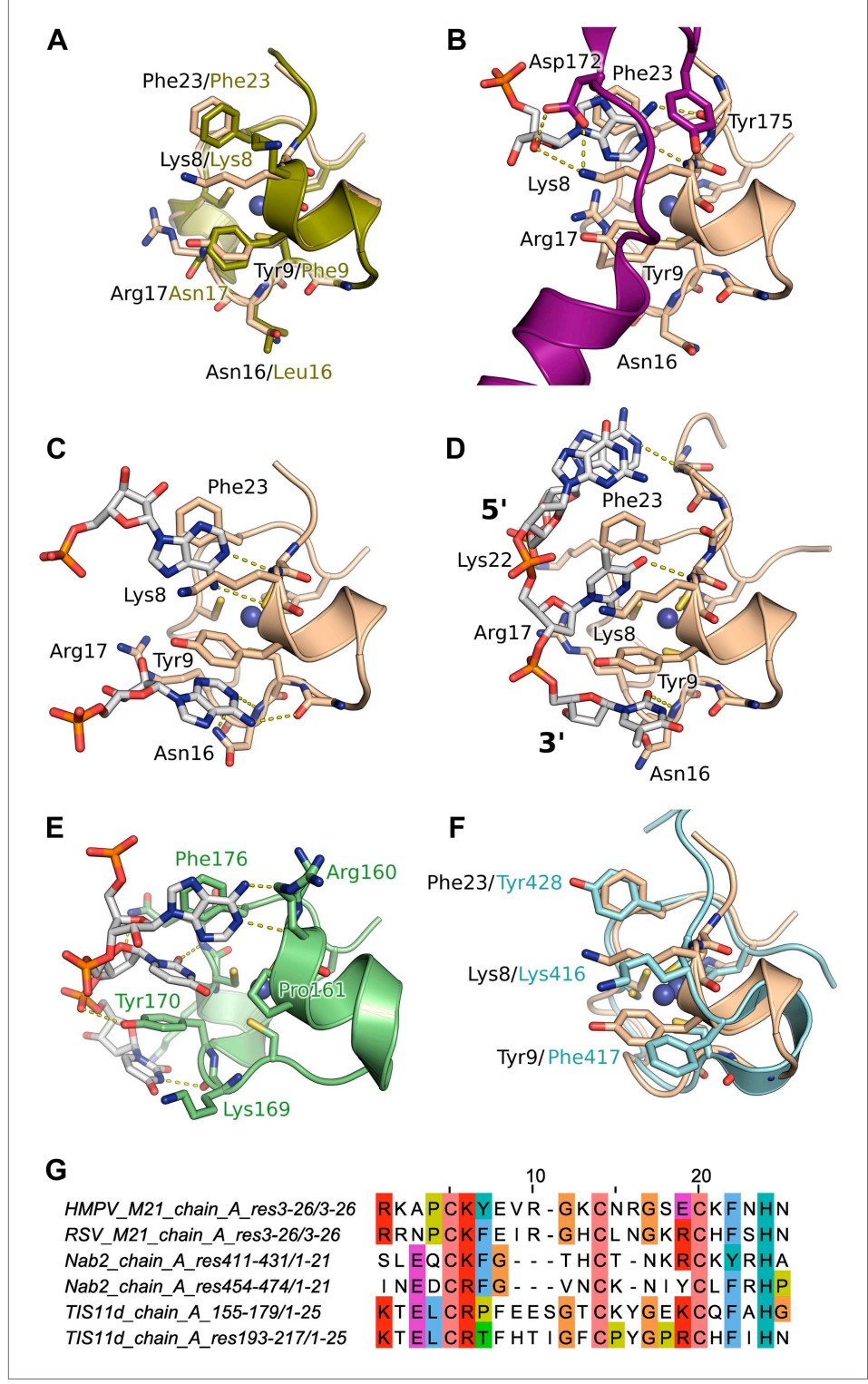

**Figure 3**. Interaction of the zinc finger of M2-1 with nucleic acids. (**A**) Superimposition of HMPV (wheat) and RSV (split pea, PDB ID 4C3B) zinc fingers, highlighting the conservation of surface residues. (**B**) Structure of adenosine monophosphate bound HMPV M2-1 zinc finger in the closed state. The C-terminal region of the core domain that folds onto the bound nucleotide is shown in purple. (**C**) Structure of adenosine monophosphate bound HMPV M2-1 zinc finger in the open state. (**D**) Structure of HMPV M2-1 bound to the DNA sequence AGTT. (**E**) NMR structure of TIS11d zinc finger 1 bound to the RNA sequence AUU (PDB ID 1RGO). (**F**) Superimposed structures of HMPV M2-1

*Figure 3. Continued on next page*

*Figure 3. Continued*

zinc finger and Nab2 zinc finger 5 (in light blue), showing the spatial conservation of the aromatic and charged residues involved in RNA binding. (**G**) Multiple sequence alignment of viral and eukaryotic zinc fingers that preferentially bind adenosine rich (HMPV and RSV M2-1, Nab2) or adenosine-uridine rich sequences (TIS11d).

The following figure supplements are available for figure 3:

**Figure supplement 1**. Effect of nucleotides on HMPV M2-1 melting profile monitored by fluorescence-based thermal shift assay.

**Figure supplement 2**. $2F_o$-$F_c$ electron density maps contoured at ~1σ, from the crystal structures of AMP and DNA bound HMPV M2-1.

base. The AMP phosphates are involved in electrostatic interactions with positively charged residues located on the core domain (*Figure 4*). Additional stability is provided in the closed state by ordering of the C-terminal loop from a neighbouring protomer within the tetramer, which folds onto the bound nucleotide (*Figure 3B*).

In the open state, exposure of the zinc finger allows binding of a second adenosine molecule that stacks with Tyr9, forms hydrogen bonds with the Asn16 backbone nitrogen and Lys14 carbonyl (*Figure 3C*). While the electron density is well-defined for the purine ring, the ribose and phosphate are disordered and show no electron density, likely due to the absence of any packing interactions (*Figure 3—figure supplement 2A*). Additionally, Asn16 side chain engages in van der Waals interactions with the adenine N1/N6. This interaction is only possible in the open state, as Asn16 is involved in H-bonding to Gln104 of the core domain of a neighbouring protomer in the closed state (*Figure 1—figure supplement 1E*). Both this observation and the folding of the C-terminal loop onto the first AMP, suggests that interdomain communication plays a role in modulating RNA recognition.

Comparison with the structure of the zinc finger from the polyadenosine binding protein Nab2 (*Brockmann et al., 2012*) indicates spatial conservation of the three main adenine-interacting residues (*Figure 3F*), suggesting similar binding modes and confirming the observed specificity. Indeed, structural alignment of the zinc finger of HMPV and RSV indicate that positions 9 and 23 tolerate both Phe and Tyr residues (*Figure 3A,G*), suggesting the exact properties of these aromatic residues don't significantly impact binding specificity.

## Binding of a DNA oligonucleotide defines the directionality of nucleic acid binding

Soaking crystals with adenine-containing dinucleotides or trinucleotides gave positive peaks in $F_o$–$F_c$ difference maps corresponding to the two binding pockets identified in the AMP soak, whereas longer RNA or DNA led to loss of diffraction (data not shown). The only exception was a soaking experiment performed with the DNA sequence AGTTA, which yielded clear density for four nucleotides (AGTT) that were bound between the core domain and the exposed (open) zinc finger of two symmetry-related M2-1 molecules. The two T bases interact in the same binding pockets as the AMP molecules (*Figure 3D*), however, the second T is partially disordered and shows density only for its phosphate and part of the nucleotide ring (*Figure 3—figure supplement 2C*), which is involved in hydrogen bonding with the M2-1 backbone (*Figure 3D*). This lack of ordering likely results from the less stable stacking of the pyrimidine ring onto Tyr9, compared to the adenosine purine ring. The two T bases are preceded by a G, which does not share any atomic contacts with the zinc finger domain, and an adenosine base which interacts with Lys22 side chain through its phosphate and engages in a hydrogen bond with Ala5 backbone nitrogen.

Importantly, this structure defined the 5′ to 3′ direction of the bound DNA, suggesting that the zinc finger will recognize an RNA motif located upstream of the core domain. Interestingly, the directionality of bound nucleic acids seems to be controlled by the position of the conserved aromatic residues. Indeed, comparisons with the (U)UAU(U) binding TIS11d tandem zinc fingers (*Hudson et al., 2004*) reveal a different directionality of the bound RNA (*Figure 3E,G*), which arises from the different position of the aromatic residue Tyr 9, which in Tis11d is a proline residue (Pro161) while Arg17 is replaced by a tyrosine (Tyr170).

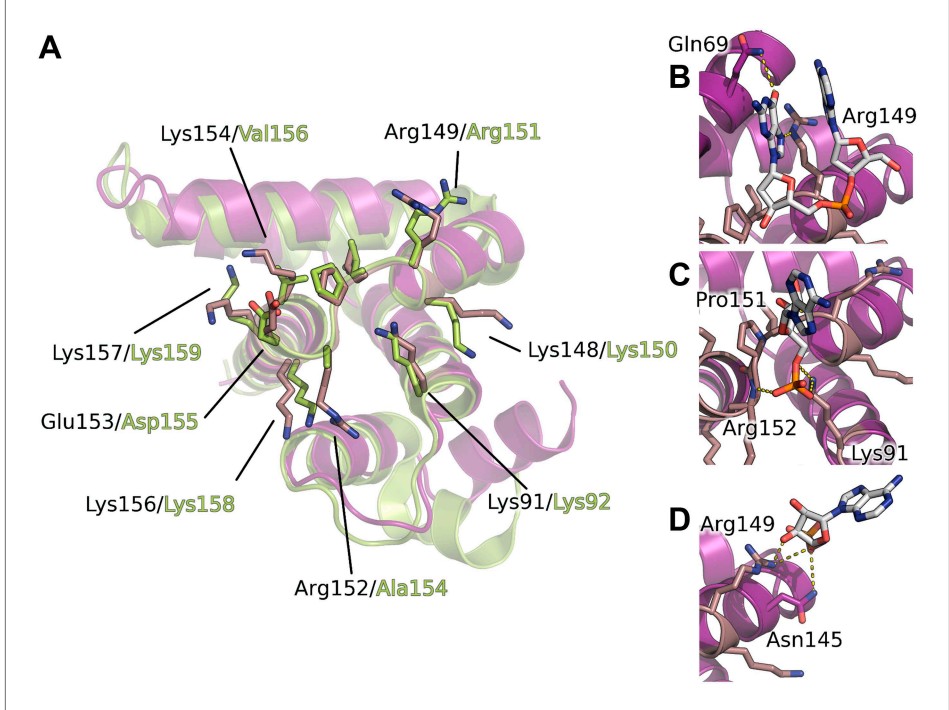

**Figure 4**. Interaction of the core domain of M2-1 with nucleic acids. (**A**) Superimposition of HMPV (purple) and RSV (lime, PDB ID 4C3D) M2-1 core domains. (**B**) HMPV M2-1 core domain bound to the DNA sequence AG. (**C** and **D**) Adenosine monophosphate bound core domains. The core domain is shown in purple cartoon representation, and RNA binding residues previously identified by NMR are coloured in brown with side chains shown in sticks.

## Comparisons between HMPV and RSV core domains indicate that the bound nucleotides map to a previously identified RNA binding surface

The core domain of RSV M2-1 has been reported to bind RNA with a 1:1 stoichiometry with significant preference for adenine rich sequences, and the binding surface has been located by NMR chemical shift perturbation experiments (*Blondot et al., 2012*). Mapping of the experimentally identified residues onto the HMPV structure indicates that the core domain binding site is conserved between the two viruses (*Figure 4A*). Our AMP and DNA bound structures show that the binding of nucleotides to the core domain is stabilized through interactions between phosphate atoms and positively charged residues which are part of the identified binding site, in particular the side chains of Lys91 and Arg149 (*Figure 4B–D*).

### Binding of viral RNA sequences to M2-1 induces the closed conformation

We performed fluorescence-based TSA titrations to determine the effect of the gene end RNA from the F gene 5′-AGUUAauuAAAAA-3′ on the M2-1 melting curves (*Figure 5—figure supplement 1*). Surprisingly, the RNA induced both positive and negative $T_m$ variations depending on the stoichiometry, which excluded the possibility of a simple binding mechanism and suggested the presence of multiple binding sites and/or different affinities for different conformations of the protein. Interestingly, studies of RSV M2-1 have suggested negative cooperativity in the binding of M2-1 to RNA (*Tanner et al., 2014*).

Next, we used the sequence of the gene end from the F gene, as well as an adenosine-rich sequence from the leader RNA (position 1 to 12, positive sense), 5′ACGCGAAAAAAU-3′, to investigate the binding of M2-1 to RNA by SAXS. Diluted M2-1 protein (~0.2 mg/ml) in 150 mM NaCl and 500 mM NDSB buffer was titrated with RNA in solution (*Figure 5A*). Unexpectedly, the addition of RNA led to a sizable increase in radius of gyration from less than 4 nm (*Table 2*) to approximately 8 nm (*Figure 5B*), likely due to the formation of nonspecific soluble aggregates. The increase was

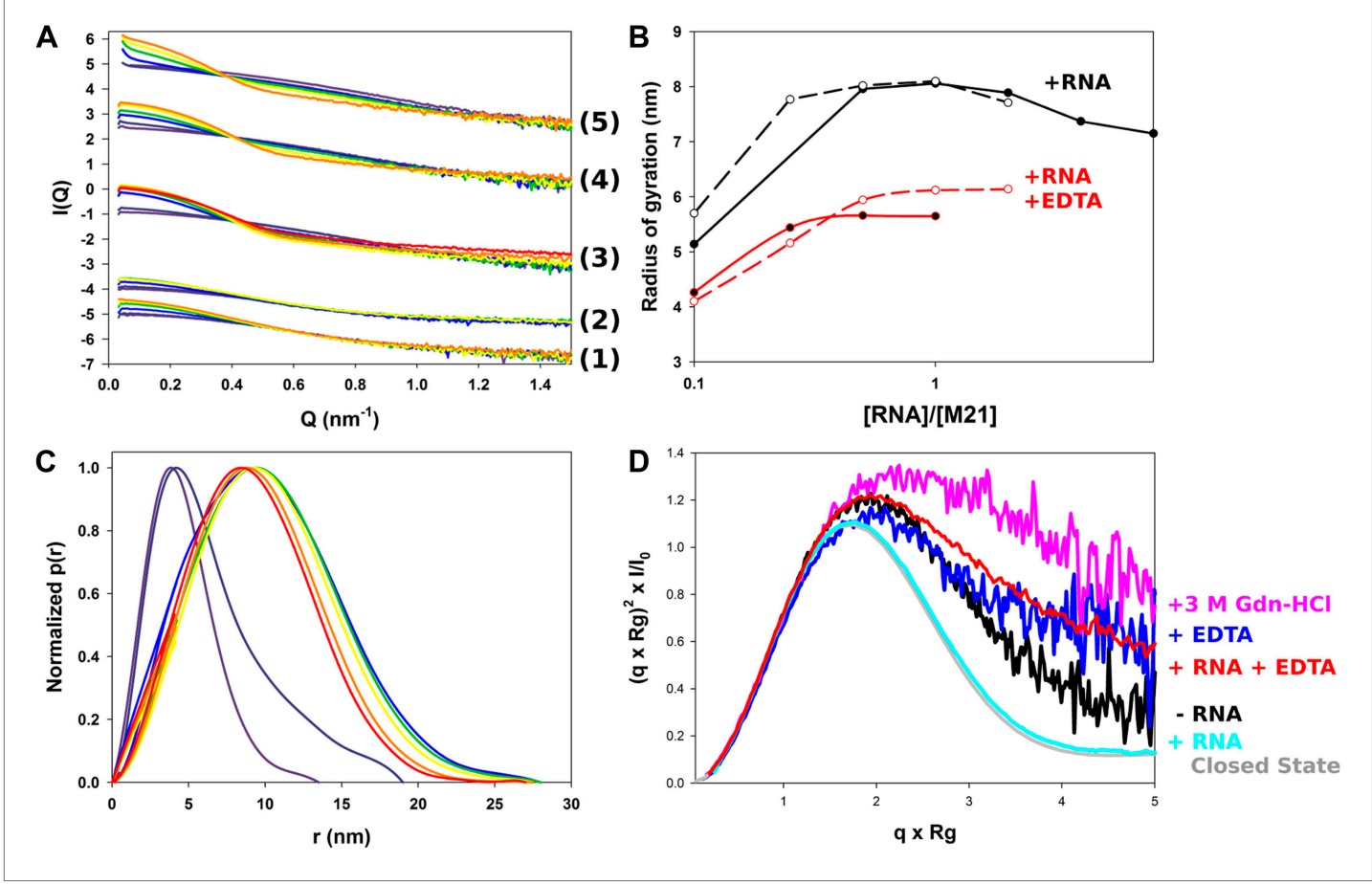

**Figure 5**. Structural characterization of M2-1/RNA complexes using SAXS. (**A**) RNA induced changes on the measured SAXS profiles by addition of leader RNA (5'ACGCGAAAAAAU-3') at 0, 1, 5, 10, 20 μM or gene end RNA (5'-AGUUAauuAAAAA-3') at 0, 1, 5, 10, 20, 40, and 80 μM, coloured from purple to red. From bottom to top, M2-1 in the presence of 20 mM Tris pH 7.5, 150 mM NaCl and 500 mM NDSB-201, 5 mM EDTA + gene end RNA (1) or leader RNA (2), M2-1 in the presence of 20 mM Tris pH 7.5, 150 mM NaCl and 500 mM NDSB-201 + gene end RNA (3) or leader RNA (4). The top curve (5) is similar to (4) but using a higher protein concentration (~1 mg/ml vs ~0.2 mg/ml). (**B**) The measured radius of gyration is plotted as a function of the molar ratio of RNA to protein concentration, indicating the formation of higher order oligomers. Titrations performed in the presence or absence of 5 mM EDTA are shown in red and in black, respectively, leader RNA titrations are in white circles and dashed lines and gene end RNA titrations are in black circles and solid lines. (**C**) Normalized pair distance distribution functions for M2-1 in 20 mM Tris pH 7.5, 150 mM NaCl and 500 mM NDSB-201 (red curve) and with increasing amounts of gene end RNA (orange to green to purple curve). (**D**) Normalized Kratky plots of M2-1 in the presence of 20 mM Tris pH 7.5, 150 mM NaCl, and 500 mM NDSB-201 (black), with 20 μM of RNA added (cyan), with 5 mM of EDTA (purple), with 5 mM of EDTA and 20 μM of RNA (red), or with 3 M Gdn-HCl (pink). Finally, the grey curve is the normalized Kratky plot calculated from the theoretical SAXS profile of an ensemble of three closed state conformers (to account for the flexibility of the disordered N-terminal side and C-terminal extension).
The following figure supplements are available for figure 5:

**Figure supplement 1**. Effect of gene end RNA (5'-AGUUAauuAAAAA-3') on HMPV M2-1 melting profile monitored by fluorescence-based thermal shift assay.

**Figure supplement 2**. Visualization of aggregated M2-1/RNA complexes by electron microscopy.

dependent upon protein concentration, and higher protein concentrations (~1 mg/ml) led to substantial aggregation, precluding accurate $R_g$ determination (*Figure 5A*, top series of curves). It is worth noting that the observed oligomerization was not due to scattering signal of the RNA samples, which were measured by SAXS and displayed $R_g$ values of less than 2 nm (data not shown). Analysis of pair distance distribution functions p(r) indicated a transition to larger species with an average $D_{max}$ of ~25 nm (*Figure 5C*).

To understand the nature of RNA-induced conformational changes in M2-1, we used dimensionless Kratky plots (**Figure 5D**). This data representation allows for a semiquantitative analysis of the degree of protein compactness independently of changes in the protein oligomeric state, by normalizing for particle size and mass (**Receveur-Brechot and Durand, 2012**). In the absence of RNA, M2-1 displayed a relatively flat peak, characteristic for decorrelated interdomain motions in modular proteins (**Bernado, 2009**), which was consistent with the presence of only 50% of closed state conformers as noted previously (**Figure 2A,I**). Addition of RNA led to a bell-shaped curve, which was almost superimposable with a theoretical plot calculated from an ensemble of closed state models, indicating that RNA binding induces M2-1 to adopt a closed conformation, coupled with concentration dependent aggregation.

## EDTA treatment of M2-1 inhibits RNA induced globularization

RNA titration of M2-1 samples incubated in the presence of 5 mM EDTA also led to an increase in radius of gyration (**Figure 5A,B**), however, its magnitude was smaller ($R_g$ = 5.7 vs $R_g$ = 8.0 nm in the absence of EDTA). This suggests that the integrity of the zinc finger is important for the RNA-induced formation of higher order oligomers.

The normalized Kratky plot obtained in the presence of EDTA revealed a higher degree of flexibility of M2-1 than in the absence of EDTA (**Figure 5D**), which is consistent with the destabilization of the closed state observed earlier (**Figure 2D,I**). The profile was intermediate between the 50% closed M2-1 measured in 500 mM NDSB and the mostly open M2-1 observed on addition of 3 M Gdn-HCl, which is shown for comparison. Interestingly, addition of RNA onto the EDTA-treated samples did not result in compaction, but rather induced a small increase in flexibility. This indicates that RNA-induced closure of M2-1 results from the simultaneous binding of the core and zinc finger domains, and that impaired binding to the zinc finger inhibits this conformational change. The moderate increase in flexibility may then result from RNA binding to the core domain only, with RNA fragments remaining unbound and disordered within the complex.

## Electron microscopy confirms that aggregates of M2-1/RNA complexes arise from assembled globular tetramers

Complexes of M2-1 and RNA were observed by electron microscopy of negatively stained samples (**Figure 5—figure supplement 2**). In the presence of RNA, M2-1 formed globular structures with diameters ranging from slightly less than 10 nm, which is consistent with the molecular dimensions of the close state tetramer ($D_{max}$ ~ 9–12 nm), up to several hundreds of nanometers. The vast majority of particles were large, linear, or branched polymers and formed of aggregated globular subunits, likely accounting for the concentration-dependent aggregation observed in solution. In contrast, negatively stained M2-1 could not be clearly identified using electron microscopy (data not shown), probably due to the combination of small size (~90 kDa) and high flexibility.

## Model of the M2-1/RNA complex

The compaction of M2-1 in presence of RNA together with the observed sequence specificity and 5′ to 3′ directionality in the zinc finger and core domains suggests a model of the M2-1/RNA complex (**Figure 6A**). In this model, a single-stranded RNA molecule can simultaneously interact with the zinc finger and core domains, resulting in saturation at a 1:1 stoichiometry, and sterically induces closure through bridging of the two domains by RNA, consistent with the EDTA preventing compaction of M2-1. The residues involved in RNA binding that were identified here for HMPV, and the ones identified earlier for RSV (**Blondot et al., 2012**; **Tanner et al., 2014**) form an almost continuous surface on the closed M2-1 structure, with dimensions compatible with the binding of a 12 to 13 nucleotide-long sequence. This is further supported by the fact that in RSV a poly-A 13-mer binds to M2-1 with nanomolar affinity while a 8-mer only displays a $K_d$ in the micromolar range (**Tanner et al., 2014**), presumably because the 8-mer is too short to bind to the core domain and zinc finger simultaneously.

In the proposed model, the adenine in position five (A5) is recognized specifically by the zinc finger, while A9 to A13 bind to the core domain, with five nucleotides covering the entire binding site. The proposed binding specificity enables a detailed interpretation of RNA binding results obtained for RSV M2-1 by **Tanner et al. (2014)**. The authors reported an affinity of ~20 nM for a poly-A 13-mer, which was decreased by only twofold on using the RSV positive SH gene end sequence (5′-AGUUAAUUAAAAA-3′), indicating that not all adenine bases are important for high affinity. Interestingly, a much larger change was observed when comparing the affinity of M2-1 for the negative F gene end (Kd ~ 1 µM), which contains a single adenosine in position five, and the poly U, C, or

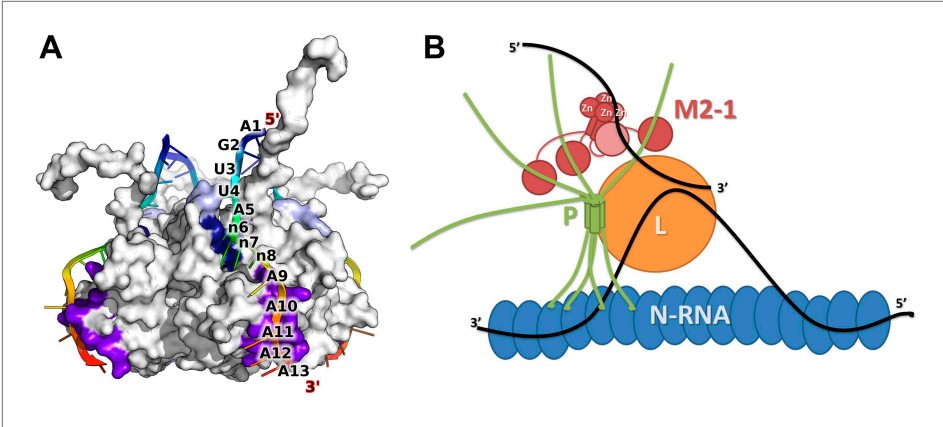

**Figure 6**. Model of M2-1 recognition of RNA and association with the viral transcription complex. (**A**) Proposed model of the recognition of the consensus gene end RNA sequence 5'-AGUUAnnnAAAAA-3' by M2-1. The model of M2-1 in its closed state is shown in white surface representation. Residues from the core domain involved in RNA binding are coloured in purple, residues from the zinc finger in dark blue, and positively charged residues identified in RSV are shown in light blue. The RNA molecule is shown in cartoon representation and coloured from blue to red (5' to 3') and the RNA sequence is mapped onto the model. (**B**) Hypothetical model of co-transcriptional recognition of viral mRNA by M2-1 in the context of the HMPV viral transcription complex. The nucleocapsid is represented in blue, and associates with RNA (black line) and with the tetrameric P protein in green. The P protein binds the viral polymerase L (orange) and the M2-1 tetramer, which is shown in red. RNA-induced conformational changes upon recognition of A-rich RNA sequences by M2-1 (gene ends or viral mRNA polyA tails) are illustrated by the closure of a single subunit bound to single-stranded RNA (light red).

The following figure supplements are available for figure 6:

**Figure supplement 1**. Comparison of functionally important surfaces identified in *pneumivirinae* M2-1 and *filoviridae* VP30 core domains, based on structural alignment by secondary structure matching (shown as cartoons in **A**).

---

G sequences (Kd >100 μM). This indicates that A5 is indeed the main determinant of M2-1 RNA binding specificity. Additional binding stability is provided by the presence of an adenine in position six (n6), consistent with a fivefold increase in affinity measured for RSV M2-1 for the positive sense SH gene end (5'-AGUUAAUUAAAAA-3') over the F gene end (5'-AGUUAUAUAAAAC-3') (*Tanner et al., 2014*), which differ by the presence of an adenine at this position, and by the length of their A tract. Nucleotides in position seven and eight (n7/8) are located in between the two RNA binding sites, consistent with them being more variable. This model suggests that the length of this variable region is important for optimal recognition. Interestingly, the leader RNA sequence used in RNA titration experiments is quite similar to the portion of the gene end sequence that is recognized by M2-1, with an adenine base located four nucleotides upstream of an $A_6$ tract.

The proposed model of M2-1 recognition of RNA can be integrated into a model of the viral transcription complex in order to understand how M2-1 may affect transcription termination in vivo (*Figure 6B*). In this model, M2-1 is tightly associated to the viral phosphoprotein P, resulting in a non-globular tetramer/tetramer complex, as has been observed in RSV (*Esperante et al., 2012*). The tetrameric P protein acts as a hub, connecting the nucleocapsid–RNA complex to the viral polymerase L and the M2-1 protein, and most probably prevents M2-1 aggregation observed in vitro. In this context, nascent viral mRNAs can be accessed by M2-1 and closure of the core domain triggered upon recognition of high affinity sequences such as gene ends and polyA tails. Because the binding of M2-1 core domain to RNA and P has been shown to be competitive due to partially overlapping binding surfaces (*Blondot et al., 2012*), the tetrameric nature of M2-1 enables specific RNA recognition, while maintaining a tight association with P.

## Discussion

The M2-1 protein functions as an antitermination factor by preventing premature transcriptional termination of long mRNAs and increases the synthesis of polycistronic read-through mRNAs by the

polymerase, at least in RSV (*Fearns and Collins, 1999*). In this study, we have solved the structure of M2-1 in the crystal and modelled it in solution, showing that M2-1 constitutes a flexible platform for recognition of specific RNA sequences. Furthermore, we found that the closed state of M2-1 is stabilized by RNA binding through simultaneous recognition of RNA by the zinc finger and core domains, a process that is likely driven by a combination of induced fit and conformational selection mechanisms (*Mackereth and Sattler, 2012*).

## Comparison with Ebola virus VP30

The VP30 protein from filoviruses is the only protein within the *Mononegavirales* that shares structural and functional similarity with the M2-1 protein from pneumoviruses. VP30 possess a CCCH zinc finger motif (residues 72 to 90) located directly upstream of an oligomerization helix (residues 94 to 112), and followed by a core domain (residues 142 to 272) that is structurally similar to M2-1 core domain (*Hartlieb et al., 2007*; *Blondot et al., 2012*). Similar to M2-1, the oligomerization and core domains are separated by a predicted disordered region. However, VP30 was found to form dimers via its core domain and is thought to form hexamers through its oligomerization domain, which contrasts with the tetramers observed for M2-1. Interestingly, mutagenesis studies of VP30 core domain have identified sets of residues that decreases both nucleocapsid protein (NP) binding and transcription activation (R179E, K180A, and K183E), or that decrease association with NP (E197A) without impairing transcription activation, that map to opposite faces of the domain (*Hartlieb et al., 2007*). Comparisons between the functional surfaces of structurally aligned M2-1 and VP30 core domains indicate that the same face of the core domain might be involved in interaction with the zinc finger and oligomerization domains in both viruses (*Figure 6—figure supplement 1*). In this scenario, mutation of VP30 R179, K180, and K183, on the face that is involved in interaction with the zinc finger and oligomerization domains, would prevent conformational changes in VP30 that are required for transcription activation, while mutation of E197 on the other face of the core domain would decrease binding to the NP protein without affecting VP30 conformation.

## The stability of the closed state is critical for M2-1 function

In solution, HMPV M2-1 samples a range of conformations between the open and closed state. This observation reconciles conflicting earlier solution and crystallographic studies obtained for RSV, where solution characterization suggested a non-globular, extended tetramer (*Esperante et al., 2011*) while the crystal structure indicated a compact, stable, intertwined tetrameric assembly (*Tanner et al., 2014*). Interestingly, mutating RSV M2-1 zinc finger residues Leu16 and Asn17 to the corresponding residues of murine pneumovirus (Ser and Arg, respectively) has identified these residues as critical to protein function (*Zhou et al., 2003*). In the RSV M2-1 crystal structure, Leu16 is involved in extensive hydrophobic contacts with the core domain, suggesting it stabilizes the closed tetramer, while Asn17 is exposed on the zinc finger surface. These residues map in HMPV to Asn16 and Arg17 (*Figure 3G*), and our structural data indicate that Asn16 is similarly involved in stabilization of the closed state of M2-1 through hydrogen bonding to Gln104. This suggests that the stability of the closed state and the fine-tuning of M2-1 dynamics are critical to antitermination. The role of Arg17 is less clear due to its aliphatic sidechain but it seems to be involved in RNA recognition, as well as intramolecular contacts with Glu20, Asn 77, and Tyr9 in the apoprotein.

## M2-1 zinc finger specifically recognizes adenosine bases and binding is modulated by the protein conformational state

Numerous studies in RSV have investigated the functional role of the CCCH zinc finger of M2-1, demonstrating its involvement in M2-1 antitermination activity (*Hardy and Wertz, 2000*; *Tang et al., 2001*; *Zhou et al., 2003*). However, the nature of this involvement has remained unclear. Here, we obtained adenosine- and DNA-bound structures of the M2-1 zinc finger and have shown its homology to RNA binding eukaryotic motifs such as Nab2 (*Brockmann et al., 2012*) or Tis11d (*Hudson et al., 2004*). Using X-ray crystallography, we showed that the M2-1 zinc finger specifically recognized two adenine bases. Binding of the first adenine base in position five is stabilized in the closed state through induced folding of the disordered C-terminal of the protein onto the bound nucleotide. The interaction with the second adenine in position six was observed only in the open state, likely due to the interaction with Asn16 which is only flipped out in the open state and thus able to contact the adenosine ring. These observations reveal intricate communication between the zinc finger and core domains, leading to a modulation of RNA binding by the protein conformational state that may be important for M2-1

function. Specifically, these results suggest that binding at position six may be less stable in the closed state than in the open state, and result in increased exposure of this nucleotide to the solvent once the M2-1 core domain closes onto the zinc finger.

## Implications for intragenic and intergenic antitermination

The preference for adenosine, the observed compaction on addition of gene end RNA and its inhibition by EDTA, and the identity of RNA binding residues (*Blondot et al., 2012*; *Tanner et al., 2014*), allow us to propose a model of gene end RNA recognition by M2-1 (*Figure 6*). The model suggests that the zinc finger recognizes A5 and A6, while the core domain binds to the poly-adenosine tract (A8 to A13). Interestingly, mutagenesis studies of the gene end sequences in RSV have shown that mutations of position five and six to nucleotides other than adenosine resulted in a strong decrease in antitermination efficiency (*Harmon et al., 2001*). This observation was surprising, given that the nucleotide in position six is not conserved within gene end sequences. However, the occurrence of an A in this position is associated in RSV with a higher affinity of the RNA for M2-1 (*Tanner et al., 2014*), and with increased antitermination efficiency at the corresponding gene junctions (*Hardy et al., 1999*). An intriguing possibility is that differential recognition of the nucleotide in position six of the gene end sequence by the zinc finger may result in exposure of n6 in the bound state, enabling the polymerase to discriminate between different gene junctions. Nucleotides n7 and n8 are located in between the two binding sites, consistent with their lack of conservation. In particular, mutations of n7 to any of the four nucleotides were found to have a minimal effect on antitermination (*Harmon et al., 2001*). In addition, the position of the A-tract relative to the upstream sequence element and a minimum length of five A bases has been shown to be critical for antitermination (*Harmon et al., 2001*).Our model is entirely consistent with these findings.

The nucleotides in position 2 to 4 (GUU) were also intolerant of change (*Harmon et al., 2001*), and are located in our model in close proximity to Arg3 and Lys4, which have been shown to have a moderate effect on RSV M2-1 RNA binding affinity (*Tanner et al., 2014*). Additionally, our DNA-bound structure suggests that M2-1 zinc finger might bind bases located in position three or four, perhaps with lower specificity as positions five and six, as no binding was observed when crystals were soaked with UMP or GMP (data not shown).

The dynamic equilibrium between the M2-1 open and closed states likely confers to the protein the flexibility required to bind non-specific RNA sequences, by allowing modulation of the distance between the A-tract and the upstream sequence element. Interestingly, in proteins with multiple binding sites linked by intrinsically disordered regions, observed higher binding rates have been explained by 'a fly casting' mechanism (*Shoemaker et al., 2000*). This suggests that the sampling of a larger volume by M2-1 through the flexible linker may result in increased RNA scanning and binding rates. It may also allow binding to both genomic RNA and nascent mRNA transcripts, contributing to intragenic antitermination, by prevention of premature dissociation of the polymerase during transcription. When the spacing between optimally recognized RNA motifs becomes sufficiently small for M2-1 to recognize them in the closed state, the protein is locked in a closed state. This feature enables M2-1 to detect gene end sequences in its environment, and presumably also poly-adenosine sequences. Conformational changes induced by simultaneous binding may be recognised by the viral polymerase L, resulting in polyadenylation and release of the mRNA transcripts. It is currently unknown whether the signal is sensed solely through the intermediate of the P protein, or involves a direct interaction between M2-1 and L as has been suggested for VP30 in filoviruses (*Groseth et al., 2009*).

Nevertheless, the tetrameric nature of both M2-1 and P, together with the reported competitive binding of P and RNA onto M2-1 core domain (*Tran et al., 2009*), suggests that the tetrameric state could enable binding of RNA simultaneously to one or two subunits without disruption of the M2-1/P interaction (*Figure 6B*). Negative cooperativity in binding RNA, as observed in RSV (*Tanner et al., 2014*), would favour this mechanism. The P protein might additionally play a role in preventing M2-1 aggregation in the presence of RNA, and act as a chaperone for M2-1 as it does for the nucleoprotein (*Leyrat et al., 2011*).

## Oligomerization of M2-1

We observed, using SAXS, that M2-1 forms higher order oligomers in a variety of conditions. In particular, we found that RNA binding induces the closed conformation of M2-1, and is coupled with aggregation into linear and branched polymers, which could be observed by electron microscopy. While

these structures may simply arise from nonspecific electrostatic interactions and are likely not relevant to M2-1 antitermination activity, the ability of M2-1 to exist in both open and closed states suggests that the protein may engage in domain swapping (*Gronenborn, 2009*). Interestingly, RSV M2-1 has been shown to mediate interactions between the viral nucleocapsids and the matrix protein (*Li et al., 2008*), and electron cryotomography of RSV virions has revealed a protein layer located between the matrix protein and ribonucleoproteins assigned to M2-1 (*Liljeroos et al., 2013*). This suggests that higher order oligomeric forms of HMPV M2-1 might be involved in viral morphogenesis and may adopt specific structures in the presence of M and viral nucleocapsids, as was recently shown for RSV, where M2-1 was found to coincide with genomic RNA and to closely follow its distribution within filamentous viral particles (*Kiss et al., 2014*).

## Conclusions

In this study, we have determined the crystal and solution structures of HMPV M2-1, unveiling a dynamic equilibrium between open and closed conformations. This defined novel molecular surfaces, accessible in the open state of M2-1, that could be utilized in structure-based drug design and aid in the development of antiviral drugs. We have established the role of the M2-1 zinc finger domain in RNA binding and revealed the homology of its RNA binding surface to the zinc fingers of the eukaryotic poly adenosine binding protein Nab2.

In addition, we have shown that RNA binding induces the closure of M2-1 due to simultaneous binding to the zinc finger and core domains, which led us to propose a model of recognition of the gene end sequences by M2-1. This model explains the conserved sequence features of viral gene ends signals and furthers our understanding of the mechanism of antitermination in the *Pneumovirinae* family of negative-strand RNA viruses. More generally, the simultaneous binding of a bipartite RNA signal across two sites that stabilises a pre-existing sub-conformation likely represents a common mechanism in multi-domain protein recognition of RNA (*Mackereth and Sattler, 2012*).

## Material and methods

### Protein cloning, expression, and purification

The HMPV M2-1 gene from strain NL1-00 (A1) was cloned into pOPINF (*Berrow et al., 2007*) for expression of M2-1 with an N-terminal His6-3C cleavage site using the ligation-independent In-Fusion system (Clontech, Mountain View, CA) following standard procedures. The integrity of the cloned construct was checked by nucleotide sequencing. The His6-3C-M2-1 construct was expressed in Rosetta2 *Escherichia coli* cells by overnight incubation under shaking at 17°C following 1 mM IPTG induction of 1 L terrific broth in presence of appropriate antibiotics. Cells were harvested by centrifugation (18°C, 20 min, 4000×*g*). Cell pellets were resuspended in 20 mM Tris, pH 7.5, 1 M NaCl and lysed by sonication. The lysate was then centrifuged for 45 min at 4°C and 50000×*g*. The supernatant was filtered and loaded on a column containing pre-equilibrated Ni-NTA Agarose (QIAGEN, Netherlands). After extensive washes, the protein was eluted in 20 mM Tris, pH 7.5, 1 M NaCl, 400 mM imidazole. Size exclusion chromatography was then carried out on a S200 column equilibrated in 20 mM Tris, pH 7.5, 1 M NaCl, or in 20 mM Tris, pH 7.5, 300 mM NaCl, 1 M NDSB-201. Optionally, The His6 tag was removed by addition of 3C protease at 4°C for 72 hr. The cleaved product was further purified through reverse Ni-NTA purification to remove His-tagged 3C protease followed by an additional gel filtration step (either in 20 mM Tris, pH 7.5, 1 M NaCl or in 20 mM Tris, pH 7.5, 300 mM NaCl, 500 mM or 1 M NDSB-201). The protein was concentrated to up to 10 mg/ml using a Millipore concentration unit (c/o 10 kDa) in presence of 500 mM or 1 M NDSB-201 in order to avoid M2-1 protein aggregation and/or precipitation at concentrations above ~1 mg/ml.

### Crystallization and data collection

Crystallization was carried out by reverse vapour diffusion using a Cartesian Technologies pipetting system (*Walter et al., 2005*). The M2-1 protein was initially prepared in 500 mM NDSB201, 5 mM DTT, 325 mM NaCl, and 10 mM tris pH7.5 and crystallized after approximately 4–14 days in 28% wt/vol Polyethylene Glycol Monomethyl Ether 2000, 0.100 M bis-Tris pH 6.5 at 20°C. These crystals led to the first apo structure (apo1). Subsequently, new crystals were produced in the same crystallization condition with protein purified in 500 mM NDSB201, 150 mM NaCl and 20 mM tris pH7.5 buffer. Optionally, crystals were soaked in mother liquor supplemented with the appropriate concentration of ligand (50 mM of AMP, UMP, CMP or GMP, 10 mM of RNA/DNA). Crystals were then frozen in liquid

nitrogen after being soaked in a mother liquor solution supplemented with 25% glycerol. Diffraction data were recorded on the I03, I04, and I24 beamlines at Diamond Light Source, Didcot, UK. All data were automatically processed by xia2 (*Winter et al., 2013*), and multiple data sets were merged where appropriate.

## Structure determination and refinement

The structure was determined by a two wavelength MAD experiment using the anomalous scattering from zinc. Data were analysed in AutoSHARP (*Vonrhein et al., 2007*). Subsequent structures were phased by molecular replacement using fragments of the first structure as search models in PHASER (*McCoy et al., 2007*). The solution was subjected to repetitive rounds of restrained refinement in PHENIX (*Adams et al., 2010*) and Autobuster (*Blanc et al., 2004*) and manual building in COOT (*Emsley et al., 2010*). TLS parameters were included in the final round of refinement. The CCP4 program suite (*Winn et al., 2011*) was used for coordinate manipulations. The structures were validated with Molprobity (*Chen et al., 2010*). Refinement statistics are given in *Table 1*, and final refined coordinates and structure factors have been deposited in the PDB with accession codes 4CS7, 4CS8, 4CS9, and 4CSA.

## Sequence- and structure-based analyses

All the structure-related figures were prepared with the PyMOL Molecular Graphics System (DeLano Scientific LLC). Protein interfaces were analyzed with the PISA webserver (*Krissinel and Henrick, 2007*). Structure-based sequence alignments were performed using PROMALS 3D (*Pei et al., 2008*).

## Small angle X-ray scattering experiments

Small angle X-ray scattering measurements for M2-1 were performed on beamline BM29 at the European Synchrotron Radiation Facility (ESRF), Grenoble, France. Data were collected at 20°C, a wavelength of 0.0995 nm and a sample-to-detector distance of 1 m. 1D scattering profiles were generated and blank subtraction was performed by the data processing pipeline available at BM29 at the ESRF. Pair distance distribution functions were calculated using the program GNOM (*Svergun, 1992*). Molecular weights were estimated based on *Rambo and Tainer (2013)*.

## Structural modelling, molecular dynamics simulations and ensemble optimization

Starting coordinates for the missing residues of M2-1 that is N-terminal His6 and 3C protease cleavage site and disordered C-terminal region (residues 168 to 187) were added in extended conformations in Modeller (*Eswar et al., 2008*). Closed and open forms of M2-1 were generated using structural alignments based on crystallographic protomers. Torsion angles were edited in VEGA ZZ (*Pedretti et al., 2002*) wherever necessary to avoid steric clashes. All classical MD simulations were performed using GROMACS 4 (*Hess et al., 2008*) and the AMBER99SB-ILDN* force field (*Best and Hummer, 2009*; *Lindorff-Larsen et al., 2010*). At the beginning of each simulation, the protein was immersed in a box of SPC/E water, with a minimum distance of 1.0 nm between protein atoms and the edges of the box. 150 mM NaCl were added using genion. Long range electrostatics were treated with the particle-mesh Ewald summation (*Essmann et al., 1995*). Bond lengths were constrained using the P-LINCS algorithm. The integration time step was 5 fs. The v-rescale thermostat and the Parrinello–Rahman barostat were used to maintain a temperature of 300 K and a pressure of 1 atm. Each system was energy minimized using 1000 steps of steepest descent and equilibrated for 200 ps with restrained protein heavy atoms. For each system, two independent production simulations were obtained by using different initial velocities. The aggregated simulation time was ~150 ns for open M2-1 and ~650 ns for close M2-1. RMSF were calculated using GROMACS routines.

Models of M2-1 of all possible combinations of open and closed states (six systems) were simulated using an atomistic coarse-grained structure-based model (SBM) (*Whitford et al., 2009*; *Noel et al., 2010*). 10,000 snapshots were extracted from each simulation, and a 1000 models from each simulation were dispatched into 10 independent ensembles of 6000 models.

For each model from each ensemble, theoretical SAXS patterns were calculated with the program CRYSOL (*Svergun et al., 1995*) and ensemble optimization fitting was performed for each of the 10 independent ensembles with GAJOE (*Bernado et al., 2007*). In order to build conformational landscapes, 50 models were selected per ensemble, resulting in 500 optimized models. Radius of gyration and $D_{max}$ values were extracted from the models and then used to calculate two-dimensional

histograms (using a binning of 15 × 15). The relative populations of conformers were counted in each optimized ensemble and averaged for each experimental SAXS profile to gain insight into the mechanism of opening and closure.

## Thermal shift assay

The thermal shift assay (ThermoFluor) was carried out in a real-time PCR machine (BioRad DNA Engine Opticon 2) where buffered solutions of protein and fluorophore (SYPRO Orange; Molecular Probes, Invitrogen, Carlsbad, CA), with and without additives, were heated in a stepwise fashion from 20°C to 99°C at a rate of 1°C/min. An appropriate volume of protein and 3 μl of SYPRO Orange (Molecular Probes, Invitrogen, Carlsbad, CA) were made up to a total assay volume of 50 μl with starting buffer (50 mM HEPES, pH 7.5, 150 mM NaCl) in white low profile thin-wall PCR plates (Abgene, Waltham, MA) sealed with microseal 'B' films (BioRad, Hercules, CA). The fluorophore was excited in the range of 470–505 nm and fluorescence emission was measured in the range of 540–700 nm every 0.5°C after a 10-s hold. The effect of nucleotides was assayed by comparing melting temperatures ($T_m$) of protein in starting buffer and in AMP, UMP, CMP, or GMP supplemented buffer. All thermal shift reactions were performed in triplicate. $T_m$, enthalpy of unfolding ($\Delta H_u$), and change in heat capacity on binding were calculated by fitting the experimental data to equations 9, 24, and 25 from *Layton and Hellinga (2010)*.

## RNA material

Synthetic RNA oligonucleotides with the sequences 5'ACGCGAAAAAAU-3' and 5'-AGUUAAUUAAAAA-3' were ordered from Thermo Scientific and dissolved in RNAse free buffers extemporaneously prior to SAXS and TSA experiments.

## Electron microscopy

Electron microscopy grids of the M2-1 and RNA mixture (ratio 1:2) were stained with 2% uranyl acetate. Images were collected using a transmission electron microscope (T12, FEI) operated at 80 kV on a CCD camera (Eagle, FEI).

## Acknowledgements

We thank Prof Ron Fouchier and Dr Bernadette van den Hoogen for providing us with a plasmid encoding HMPV M2-1, Dr Alistair Siebert for help with electron microscopy, Dr Luigi De Colibus and Prof David Stuart for helpful discussions. The research leading to these results has received funding from the European Union Seventh Framework Programme (FP7/2007-2013) under SILVER grant agreement n° 260644. Administrative support came from the Wellcome Trust Core award (090532/Z/09/Z), MR was supported by a Wellcome Trust Studentship (099667/Z/12/Z), and JTH by the Academy of Finland (130750 and 218080). The OPIC electron microscopy facility was founded by a Wellcome Trust JIF award (060208/Z/00/Z) and is supported by a WT equipment grant (093305/Z/10/Z). The work presented here made use of the High Performance Computing facility IRIDIS provided by the EPSRC funded Centre for Innovation (EP/K000144/1 and EP/K000136/1) which is owned and operated by the e-Infrastructure South Consortium formed by the universities of Bristol, Oxford, Southampton and UCL in partnership with STFC's Rutherford Appleton Laboratory. The authors thank Diamond Light Source for beamtime (proposal mx8423), and the staff of beamlines I03, I04, and I24 for assistance with crystal testing and data collection, as well as the ESRF for SAXS beamtime, and the staff of beamline BM29 for assistance with SAXS data collection.

# Additional information

## Funding

| Funder | Grant reference number | Author |
| --- | --- | --- |
| European Commission | FP7/2007-2013 SILVER 260644 | Cedric Leyrat, Jonathan M Grimes |
| Wellcome Trust | 090532/Z/09/Z | Cedric Leyrat, Max Renner, Karl Harlos, Juha T Huiskonen, Jonathan M Grimes |

| Funder | Grant reference number | Author |
| --- | --- | --- |
| Wellcome Trust | 060208/Z/00/Z and 093305/Z/10/Z | Juha T Huiskonen |
| Wellcome Trust | 099667/Z/12/Z | Max Renner |
| Acadamy of Finland | 130750 | Juha T Huiskonen |
| Acadamy of Finland | 218080 | Juha T Huiskonen |

The funders had no role in study design, data collection and interpretation, or the decision to submit the work for publication.

### Author contributions

CL, JMG, Conception and design, Acquisition of data, Analysis and interpretation of data, Drafting or revising the article; MR, JTH, Acquisition of data, Analysis and interpretation of data, Drafting or revising the article; KH, Karl helped the 2 first authors grow and prepare the crystals for data collection, as well as collecting the crystallographic data., Acquisition of data, Analysis and interpretation of data, Contributed unpublished essential data or reagents

# Additional files

## Major datasets

The following datasets were generated:

| Author(s) | Year | Dataset title | Dataset ID and/or URL | Database, license, and accessibility information |
| --- | --- | --- | --- | --- |
| Leyrat C, Renner M, Harlos K, Huiskonen JT, Grimes JM | 2014 | Crystal structure of the asymmetric human metapneumovirus M2-1 tetramer, form 1 | 4CS7; http://www.rcsb.org/pdb/explore/explore.do?structureId=4CS7 | Publicly available at the Protein Data Bank (http://www.rcsb.org/pdb/) |
| Leyrat C, Renner M, Harlos K, Huiskonen JT, Grimes JM | 2014 | Crystal structure of the asymmetric human metapneumovirus M2-1 tetramer, form 2 | 4CS8; http://www.rcsb.org/pdb/explore/explore.do?structureId=4CS8 | Publicly available at the Protein Data Bank (http://www.rcsb.org/pdb/) |
| Leyrat C, Renner M, Harlos K, Huiskonen JT, Grimes JM | 2014 | Crystal structure of the asymmetric human metapneumovirus M2-1 tetramer bound to adenosine monophosphate | 4CS9; http://www.rcsb.org/pdb/explore/explore.do?structureId=4CS9 | Publicly available at the Protein Data Bank (http://www.rcsb.org/pdb/) |
| Leyrat C, Renner M, Harlos K, Huiskonen JT, Grimes JM | 2014 | Crystal structure of the asymmetric human metapneumovirus M2-1 tetramer bound to a DNA 4-mer | 4CSA; http://www.rcsb.org/pdb/explore/explore.do?structureId=4CSA | Publicly available at the Protein Data Bank (http://www.rcsb.org/pdb/) |

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
