## [Decision Letter]

Thank you for sending your work entitled “Drastic changes in conformational dynamics of the antiterminator M2-1 regulate transcription efficiency in *Pneumovirinae* for consideration at *eLife*. Your article has been favorably evaluated by a Senior editor and 3 reviewers, one of whom is a member of our Board of Reviewing Editors, and one of whom, Gino Cingolani, has agreed to reveal his identity.

The Reviewing editor and the other reviewers discussed their comments before we reached this decision, and the Reviewing editor has assembled the following comments to help you prepare a revised submission:

All reviewers agreed that your structural and biophysical characterization of the dynamics and the different conformational states of the viral antiterminator M2-1 Protein is of high interest to the scientific community and of high technical standard. The following questions, however, should be addressed:

Results:

1) There is some doubt about the biological relevance of the observed higher oligomers seen in the EM images. Could this be simply aggregation? Especially Figure 5 may just show aggregates.

It is a well known phenomenon that titration of DNA (or RNA) into solutions of transcription factors (or RNA binding proteins) often leads to the formation of aggregates by non-specific binding of protein to the oligonucleotides by electrostatic interactions. Therefore, for high resolution structural investigations the titrations are usually carried out in the opposite way: Protein into a solution of DNA (or RNA) to prevent this aggregation phenomenon. Have such reverse titrations been carried out? Such a reverse titration can answer the question if this is formation of specific oligomers or simply aggregation. This question should be answered or the aggregation/higher oligomer formation part be removed.

2) Are the authors sure that addition of the Guanidinium Hydrochloride does not unfold for example the zinc finger domain? The addition of EDTA which unfolds the zinc finger domain seems to populate similar regions in the plot as the addition of 2M Gd Hcl. This issue could in principle be easily investigated by measuring CD spectra at the same condition as the SAXS data which would either show that all secondary structure elements are preserved or not.

Discussion:

3) It is not clear at all why the protein is a tetramer. It would be good if the authors could discuss this issue more. For an antitermination protein a tetrameric state would not be necessary. As a coating protein as suggested by the authors a tetrameric state would be more important, but that would be a new function and not connected to antitermination. Some more insight into this issue would be very useful.

Specifically: Hartlieb et al (2006) showed that Ebola virus VP30, whose core structure is very similar to M2.1 is involved in both transcription regulation via RNA binding and nucleocapsid interaction. Hartlieb identified three sets of mutants, one has no effect and the third reduces nucleocapsid interaction. How would these mutants affect the open or closed form? RNA binding?

4) The authors cite, but don't really discuss Tanner's symmetric structure of M2-1. How can M2-1 conformational dynamics revealed in this study be used to decipher and reconcile Tanner's perfectly symmetric structure? Tanner's structure was crystallized at pH 10. Have the authors checked by SAXS (or MD simulation) if basic pH can induce a conformational change consistent with M2-1 symmetrization? How do they explain a fully symmetric conformation of M2-1?

5) Following up on point #4, could asymmetry seen in this paper be 'enhanced' by high salt and/or NDSB-201 used during purification? Interestingly, Tanner et al express M2-1 as a GST-fusion and purify it under native conditions.

---

## [Author Response]

*Results*:

*1) There is some doubt about the biological relevance of the observed higher oligomers seen in the EM images. Could this be simply aggregation? Especially*
Figure 5
*may just show aggregates*.

*It is a well known phenomenon that titration of DNA (or RNA) into solutions of transcription factors (or RNA binding proteins) often leads to the formation of aggregates by non-specific binding of protein to the oligonucleotides by electrostatic interactions. Therefore, for high resolution structural investigations the titrations are usually carried out in the opposite way: Protein into a solution of DNA (or RNA) to prevent this aggregation phenomenon. Have such reverse titrations been carried out? Such a reverse titration can answer the question if this is formation of specific oligomers or simply aggregation. This question should be answered or the aggregation/higher oligomer formation part be removed*.

We agree with the comments concerning the biological relevance of these observations and concede that we should have been more explicit in the Results and Discussion in stating that the aggregation observed by SAXS and EM in presence of RNA reflects biophysical, rather than biological properties of the protein. Indeed, as pointed out by the reviewers, they may arise from non-specific electrostatic interactions.

The rationale for including the EM observations of M2-1 aggregates is technical rather than biological, as it confirms that these assemble from globular units, as inferred from solution SAXS data. The SAXS reveals that even though aggregation occurs, by analysing the normalized Kratky plots, we are still able to extract information on the open and closed state of M2-1 in the presence of RNA. Normalized Kratky plots represent a reliable, model-free analysis that is frequently used in polymer science but was only recently popularized for the analysis of protein flexibility (Receveur-Bréchot et al., 2012). This type of analysis thus is a facet of BioSAXS that is not commonly exploited, and data showing the formation of soluble aggregates are often discarded without further analysis. For this reason, we believe it is useful to confirm the SAXS analysis with an additional technique such as EM.

Since M2-1 functions in complex with N, P and L within the viral transcription/replication complex, the ability of M2-1 to associate with RNA and/or with neighboring M2-1 tetramers is tightly controlled in vivo. Specifically, association of M2-1 with tetrameric P is likely to prevent the formation of any atomic contacts between M2-1 tetramers and thus inhibit M2-1 aggregation. Additionally, genomic RNA is encapsidated by N and likely is not accessible to M2-1 because of steric hindrance from P and L. Consequently, only transcribed viral mRNA will be available for binding M2-1 and thus M2-1 will never exist in the presence of an excess of RNA. Thus carrying out a reverse titration of M2-1 into a solution of RNA would then also be of doubtful biological relevance since, in the context of the viral lifecycle, M2-1 will not be in excess of RNA and N,P and L are not present. We have produced a scheme illustrating how M2-1 may function in the presence of N, P and L, which has been included as a new panel in Figure 6.

To clarify this issue of biological relevance, the titles and contents of the paragraphs describing M2-1 aggregation in the Results and Discussion have been modified, the EM observations have been moved to a new figure supplement, and it is now clearly stated in the text that M2-1 aggregation is not relevant to M2-1 antitermination activity.

*2) Are the authors sure that addition of the Guanidinium Hydrochloride does not unfold for example the zinc finger domain? The addition of EDTA which unfolds the zinc finger domain seems to populate similar regions in the plot as the addition of 2M Gd Hcl. This issue could in principle be easily investigated by measuring CD spectra at the same condition as the SAXS data which would either show that all secondary structure elements are preserved or not*.

The reviewers point out that Gdn-HCl may unfold the Zn finger domain of M2-1, which indeed is a possibility that cannot be completely ruled out on the basis of the SAXS data. Our ensemble analysis allows us to conclude that Gdn-HCl doesn't cause tetramer dissociation, or unfolding of the tetramerization helix or core domains, as this would result in larger increases in radius of gyration, and in the case of tetramer dissociation, changes in the calculated molecular weight. For the same reason, it is likely that the zinc finger doesn't adopt completely extended conformations in the presence of moderate amounts of Gdn-HCl. However, in the case of the Zn-finger domain, the expected changes on the scattering profile would be small and thus we can't exclude this possibility. This has now been clearly stated in the text.

As suggested by the reviewers, changes in M2-1 secondary structure could in principle be monitored via circular dichroism. The analysis is in practice complicated by several factors. (1) In the case of M2-1, the equilibrium between open and closed states affects the stability of α1 in the zinc finger domain, and causes unfolding of the first alpha helix of the core domain (α3). This is in agreement with the crystal structures, in which higher B factors and lower quality electron density for the zinc finger are observed in the open state. Furthermore, unfolding of α3 is accounted for in our MD simulations and ensemble analysis of the SAXS data. This is also consistent with the fact that in RSV M2-1, α3 is present in the crystal structures reported by [53], however, adopts unfolded conformations in the NMR structure of the core domain from [8]. These observations would make it difficult to assign any changes in the CD spectra to a particular region of the protein. (2) Measuring CD spectra of M2-1 in the presence of Gdn-HCl, NDSB-201, Tris and NaCl is challenging due to the heavy absorption of these compounds in the far UV region. This could in principle be overcome by using a shorter path length, however a very high concentration of M2-1 would be needed and this is very difficult to reach. (3) It is well known that low concentrations of Gdn-HCl can induce molten globule states which retain some degree of compactness but may have a lower secondary structure content than the native protein.

*Discussion*:

*3) It is not clear at all why the protein is a tetramer. It would be good if the authors could discuss this issue more. For an antitermination protein a tetrameric state would not be necessary. As a coating protein as suggested by the authors a tetrameric state would be more important, but that would be a new function and not connected to antitermination. Some more insight into this issue would be very useful*.

*Specifically: Hartlieb et al (2006) showed that Ebola virus VP30, whose core structure is very similar to M2.1 is involved in both transcription regulation via RNA binding and nucleocapsid interaction. Hartlieb identified three sets of mutants, one has no effect and the third reduces nucleocapsid interaction*. *How would these mutants affect the open or closed form? RNA binding?*

We have modified the text to include a more in-depth discussion of the role of M2-1 tetramerization, as well as a new panel in Figure 6 to illustrate a model for the interaction of M2-1 with the viral transcription complex. We believe that the requirement for a tetrameric M2-1 results from its function in anti-termination, and allows optimal binding to tetrameric P. [54], have shown that there is a competition between RNA and P binding to the M2-1 core domains, which suggests that the formation of a tetramer/tetramer complex between M2-1 and P enables M2-1 to recognize and close onto a single stranded viral mRNA without dissociating from the viral transcription complex.

The reviewers pointed out that comparisons between HMPV M2-1 and EBOV VP30 are missing from our study. We have run disorder predictions on VP30, which suggest that the protein might share a similar modular organization as M2-1, with additional disordered regions located at the N terminus and C terminus, as well as a much longer disordered region connecting the oligomerization domain with the core domain. The VP30 protein has been suggested to form hexamers via its oligomerization domain (residues 94 to112), and dimers via its core domain. Hartlieb et al. have identified mutants located on two opposite faces of the protein, both of which affected the ability of VP30 to co-immunoprecipitate the NP protein. One set of mutations (R179, K180, K183) also affected VP30 transcription activation. It is interesting to speculate on the effect these mutations could have on VP30 conformation and RNA binding and we have generated a figure to map and compare identified functional surfaces on M2-1 and VP30, as well as a new paragraph in the discussion*.*

*4) The authors cite, but don't really discuss Tanner's symmetric structure of M2-1*. *How can M2-1 conformational dynamics revealed in this study be used to decipher and reconcile Tanner's perfectly symmetric structure? Tanner's structure was crystallized at pH 10. Have the authors checked by SAXS (or MD simulation) if basic pH can induce a conformational change consistent with M2-1 symmetrization? How do they explain a fully symmetric conformation of M2-1?*

As pointed out by the reviewers, a discussion of the factors leading to symmetric versus asymmetric crystal structures of M2-1 is missing from our study. Interestingly, WT RSV M2-1 was crystallized at pH 10 in the presence of 0.1 M of 3-(cyclohexylamino)-1-propanesulfonic acid (CAPS), and mutant RSV M2-1 (S58DS61D) crystallized at pH 8.5 and in the presence of 0.2 M trimethylamine N-oxide (TMAO), which is a well known stabilizer of protein folded states. The conformation of M2-1 may be further affected by the use of high pH, as the interface is rich in positively charged residues with a total of eight arginine or lysine residues. Most of these residues are involved in salt bridges and likely possess high pKa values, however, Lys19 and Arg38 are primarily involved in hydrogen bonds with neutral residues (Ala73 backbone oxygen and Asn40, Asn44 side chains), which may result in deprotonation at pH 10 and overstabilization of the closed state.

Regarding the solution structure of HMPV M2-1, our ensemble analysis shows that in the buffer conditions that we used for crystallization (150 mM NaCl and 500 mM NDSB-201), 50% of the molecules adopt a closed, symmetric conformation, 30% display dissociation of a single core domain (which is the state seen in the crystal structures), while the remaining molecules are in more open states, mostly with two core domains dissociated (15%), as can be seen in Figure 2, white histogram bars. Given this distribution, one would expect the protein to have crystallized in its closed, symmetric state rather than in the intermediately open state observed in HMPV M2-1 crystals. However, it appears that this intermediate state was able to generate a stable crystal lattice, resulting in an equilibrium shift. These observations highlight the importance of using solution techniques to assess whether protein conformations seen in crystalline states are consistent with the structure in solution.

*5) Following up on point #4, could asymmetry seen in this paper be 'enhanced' by high salt and/or NDSB-201 used during purification? Interestingly, Tanner et al express M2-1 as a GST-fusion and purify it under native conditions*.

The discussion regarding the effect of salt and NDSB-201 on the HMPV M2-1 structure is interesting and we have included additional SAXS data as a new supplement of Figure 2. Data was measured in the presence of 600 mM, 1.15 M or 3 M NaCl and could not be fitted by our ensembles due to the presence of a small proportion of soluble aggregates/higher order oligomers. However, we were able to use normalized Kratky plots to assess the flexibility of M2-1 in these conditions by comparing them to the Kratky plots obtained in the presence of NDSB-201, Gdn-HCl or with a theoretical closed state ensemble of models. The data shows that the conformational equilibrium of M2-1 in the presence of 600 mM to 3 M NaCl is relatively similar to that observed with 500 mM NDSB-201, showing only a slight compaction in 3 M NaCl, which likely results from an increased hydrophobic effect. This additionally suggests that the main effect of NDSB-201 on the conformational equilibrium of M2-1 may be to prevent aggregation.